# Broiler Chicken Cecal Microbiome and Poultry Farming Productivity: A Meta-Analysis

**DOI:** 10.3390/microorganisms12040747

**Published:** 2024-04-07

**Authors:** Dmitry Deryabin, Christina Lazebnik, Ludmila Vlasenko, Ilshat Karimov, Dianna Kosyan, Alexander Zatevalov, Galimzhan Duskaev

**Affiliations:** 1Federal Research Centre of Biological Systems and Agrotechnologies of the Russian Academy of Sciences, January 9 Street, 29, 460000 Orenburg, Russia; christinakondrashova94@yandex.ru (C.L.); lv.efremova@yandex.ru (L.V.); kosyan.diana@mail.ru (D.K.); gduskaev@mail.ru (G.D.); 2Orenburg State Medical University of the Ministry of Health of Russia, Sovetskaya Street, 6, 460014 Orenburg, Russia; ifkarimov@yandex.ru; 3G.N. Gabrichevsky Research Institute for Epidemiology and Microbiology, Admiral Makarov Street, 10, 125212 Moscow, Russia; zatevalov@mail.ru

**Keywords:** broiler chickens, cecal microbiome, microbiome patterns, poultry farming productivity

## Abstract

The cecal microbial community plays an important role in chicken growth and development via effective feed conversion and essential metabolite production. The aim of this study was to define the microbial community’s variants in chickens’ ceca and to explore the most significant association between the microbiome compositions and poultry farming productivity. The meta-analysis included original data from 8 control broiler chicken groups fed with a standard basic diet and 32 experimental groups supplemented with various feed additives. Standard Illumina 16S-RNA gene sequencing technology was used to characterize the chicken cecal microbiome. Zootechnical data sets integrated with the European Production Effectiveness Factor (EPEF) were collected. Analysis of the bacterial taxa abundance and co-occurrence in chicken cecal microbiomes revealed two alternative patterns: *Bacteroidota*-dominated with decreased alpha biodiversity; and *Bacillota*-enriched, which included the *Actinomycetota*, *Cyanobacteriota* and *Thermodesulfobacteriota* phyla members, with increased biodiversity indices. *Bacillota*-enriched microbiome groups showed elevated total feed intake (especially due to the starter feed intake) and final body weight, and high EPEF values, while *Bacteroidota*-dominated microbiomes were negatively associated with poultry farming productivity. The meta-analysis results lay the basis for the development of chicken growth-promoting feed supplementations, aimed at the stimulation of beneficial and inhibition of harmful bacterial patterns, where relevant metagenomic data can be a tool for their control and selection.

## 1. Introduction

Broiler chickens are the most widely distributed bird in the world. Moreover, chicken meat production worldwide exceeds 100 million tons annually thus making broiler chicken the most accessible animal protein for the general human population [1]. Therefore, further improvements in the efficiency of chicken breeding and feeding are becoming increasingly important for both food security and economic reasons.

Current consensus indicates that an optimal gut microbial community present throughout the gastrointestinal tract is a central factor determining chicken health and influencing poultry farming productivity. The normal chicken gut microbiota plays an important role in the protection against pathogenic bacteria, as well as immune system maturation and tuning, intestinal development, and effective feed conversion [2]. Additionally, some bacterial metabolites such as short-chain fatty acids, essential vitamins and amino acids are significant for effective chicken growth and development [3].

Among the different segments of the chicken gut, the most densely microbial community was found in the ceca, a pair of blind-ended sacs that open into the large intestine [4], and most of the available information about the chicken gut microbiota has been focused on the cecal microbiome. There are several systematic reviews on the chicken cecal microbiome based on Sanger technology [5], 454 pyrosequencing [6], and Illumina sequencing platforms [7]. The data considered revealed the predominant phyla and genera and characterized the alpha diversity of the microbiome, which creates a baseline for future research of microbial community in the chicken cecum.

There is some evidence that the cecal microbiome composition and diversity may vary between chickens kept in different types of indoor cage and cage-free systems [8], the use of different types of litter substrates [9], and especially different diet supplementation [10]. In turn, varieties in the cecal microbiome influenced nutrient metabolism [11], including fat metabolism and fat deposition in broiler chickens [12], and were significant for growth promotion and individual differences in chicken body weight [13]. However, a comprehensive analysis of the relationship between cecal microbiome composition and chicken productivity is still unavailable.

In a series of our previous studies published in international [14,15,16,17,18] and local [19,20,21,22,23] scientific journals, we used standard Illumina 16S-RNA gene sequencing technology in the context of the effects of different feeding regimes on the microbiome, feed efficiency, and growth performance indices in broiler chickens. General changes in cecal microbiome composition and zootechnical parameters were shown, but the relationship between cecal microbiota composition and diversity against poultry farming productivity was constrained by a small sample size in each series.

For this reason, we turned our attention to meta-analysis, a method of statistically combining the results of published studies that have assessed the same experimental data [24]. This approach has now become very popular because it allows for the resolution of disagreements among multiple experiments in animal nutrition and the identification of interdependencies that would not normally be visible in a single original investigation [25].

The aims of this meta-analysis study were to define the composition of the microbial communities within chickens’ ceca and to explore the most significant association between the microbiome variants and poultry farming productivity.

## 2. Materials and Methods

### 2.1. Experimental Design and Data Collection

Results of ten experimental series performed during the period of 2018–2023 [14,15,16,17,18,19,20,21,22,23] were included in the meta-analysis (Figure 1). The inclusion criteria were: (i) the study implementation to be in accordance with the ethical statement and the principles of good laboratory practice; (ii) use of broiler chickens aged from 7 to 42 days (experiment duration of 35 days); (iii) keeping birds on a standard basic diet supplemented with various additives; (iv) standard recording of zootechnical data integrated with the European Production Efficiency Factor (EPEF); (v) standard Illumina 16S-RNA gene sequencing technology to characterize the cecal microbiome. These criteria made it possible to ensure good comparability of data from the original experimental series and were a condition for further correct statistical analysis.

Data collection details are described as follows. The study protocols were approved by the Animal Ethics Committee of the Federal Scientific Center for Biological Systems and Agro-Technologies of the Russian Academy of Sciences, based on the European Convention for the Protection of Vertebrate Animals used for Experimental and other Scientific Purposes (18 March 1986). Seven-day-old broiler chickens were randomly divided into equal replicates and fed with the basic diets intended for the starting period (7–28 days), and growing period (29–42 days). The control group in each series received the basic diet only, while the experimental groups were fed a diet supplemented with *Oak Bark* extract [14,15,19], *Bifidobacterium*-containing or *Bacillus*-containing probiotic additives [15,16], the antibiotic chlortetracycline [14,20,21], or chemically synthesized analogues of the following plant-derived molecules: gamma-lactone, coumarin derivative, vanillin/vanylic acid, resorcinol derivative, or quercetin [16,17,18,20,21,22,23]. In the current context, these supplements affected the cecal microbiome composition and chicken productivity, providing variability in the data for the meta-analysis. A total of 40 groups (8 control and 32 experimental) meeting these criteria were included in the study.

During the experiments, the starter and grower feeds intake, feed conversion ratio per kg of body weight, average daily gain, absolute gain, and chicken survival rate were recorded in each of the control and experimental groups. Based on the collected zootechnical data, the effectiveness of poultry farming productivity was assessed by an EPEF value calculated as (average weight gain per day × survival rate)/feed conversion × 10.

At the end of the experiments, randomly selected chickens from each group were humanely euthanized and aseptically dissected, after which the contents of each cecum were massaged into sterile cryogenic tubes, snap frozen in dry ice, and stored at −80 °C as previously described [18].

The experimental options were the use of the Smena-8 cross (Center for Genetics and Selection “Smena”—filial of Federal Scientific Center “All-Russian Research and Technological Institute of Poultry” of Russian Academy of Sciences, Moscow region, Russia) in three experimental series [14,15,19] or the Arbor Acres cross (Aviagen LLC, Tula, Russia) in seven experimental series [16,17,18,19,20,21,22,23], as well as variations in the group size in the range of 15–30 animals, which did not have significant implications for the study design and did not interfere with the meta-analysis procedure.

### 2.2. DNA Extraction, Sequencing, and Data Processing

Total DNA extraction from 40% of the cecal samples was performed using the Fast DNA SPIN Kit for Feces (MP Biomedicals Inc., Irvine, CA, USA), and 10% using the QIAamp Fast DNA Stool Mini Kit (Qiagen GmbH, Hilden, Germany) according to the manufacturer’s instructions. Half of the samples (50%) were extracted using the phenol–chloroform method. The DNA concentration was assessed using a Qubit 4 fluorimeter (Life Technologies, Carlsbad, CA, USA) using the dsDNA high-sensitivity assay kit. The DNA quality was evaluated by 1% agarose gel electrophoresis.

The cecal microbiome analysis in all the experimental series was standardly focused on the V3-V4 region of the 16S rRNA gene. The Illumina 16S Metagenomic Sequencing Library Preparation two-stage protocol (Part #15044223, Rev. B) was implemented. The targeted variable regions were amplified using the forward S-D-Bact-0341-b-S-17 (CCTACGGGNGGCWGCAG) and the reverse S-D-Bact-0785-a-A-21 (GACTACHVGGGTATCTAATCC) primers containing the overlapping region of the Illumina sequencing primers. The prepared DNA libraries were validated by real-time PCR on CFX Connect (BioRad, Hercules, CA, USA). The libraries were purified using solid-phase reversible immobilization on Agencourt AM Pure XP beads (Beckman Coulter, Brea, CA, USA). The quality of the DNA libraries was analyzed using the QIAxcel Advanced system (Qiagen GmbH, Hilden, Germany). Finally, high-throughput paired-end 2 × 250 bp sequencing was performed on the Illumina MiSeq platform with the V.2 reagent kit (Illumina Inc., San Diego, CA, USA) for 500 cycles.

Raw sequencing data processing was performed using the USEARCH V.10.0.240 software and included the merging of reads into contigs, filtering of contigs by length (at least 420 bp) and quality (maxee 1.0), chimera deletion, dereplication, and clustering into separate operational taxonomic units (OTU). The OTUs were assigned by the RDP classifier [26] against the SILVA 16S rRNA database (http://www.arb-silva.de/) accessed on 1 March 2021. Since the 2021 meeting of the International Committee on Taxonomy of Prokaryotes revised the nomenclature [27], publication data for 2018–2020 were further reclassified, ensuring comparability of legitimate taxa across all the experimental series included in the meta-analysis. The final OTUs taxonomic affiliation at the phylum, class, order, family, and genus level was determined according to the NCBI taxonomy browser [28] and the Genome taxonomy database (GTDB) (https://gtdb.ecogenomic.org/) accessed on 6 November 2023. (Figure 1).

### 2.3. Statistical Analysis

Basic data analysis, statistics, and visualization procedures were performed using the Statistica 10.0 software package (TIBCO Software Inc., Palo Alto, CA, USA). 

The relative abundances of the bacterial phyla found in each group were represented by column charts. Co-occurrence of different taxa in microbiomes was assessed by Pearson’s multiple correlation coefficients using the *p* < 0.001 criteria. A multivariate statistical method called principal component analysis (PCA) was used to evaluate typical microbiome patterns. Microbiome alpha diversity was estimated by the Shannon and Simpson (1-D) indexes calculated using PAST V. 4.06 software. Alpha diversity indices distribution between alternative microbiome patterns groups were assessed using one-way analysis of variance. The influence of the microbiome characteristics on the zootechnical data sets was represented by the R^2^ coefficient of determination. The distribution of zootechnical parameters between groups of broiler chickens with different cecal microbiome patterns was assessed using the Mann–Whitney U test. Canonical correlation analysis was used to examine the relationship between the microbiome characteristics and EPEF values, where *p* values < 0.05 were considered significant.

## 3. Results

### 3.1. Chicken Cecal Microbiome Composition

Analysis of a subset of 16S rRNA gene reads affiliated 99.61% OTUs with eleven relevant phyla, while 0.39% OTUs could not be correctly characterized and were assessed as *unclassified Bacteria*. At this taxonomic level, the dominant phyla of all samples were both *Bacillota* and *Bacteroidota*, found in 100% of the control and experimental groups and varying in relative abundance in the cecal bacterial community from 28.66 to 90.12% and from 4.18 to 69.09%, respectively. The *Actinomycetota* phylum was also found in all groups, but its relative abundance was significantly lower—from 0.01 to 4.65%. Resident members of the chicken cecal microbiome also included the phyla *Cyanobacteriota*, *Pseudomonadota* and *Thermodesulfobacteriota*, present in at least three-quarters of the analyzed groups. Phylum *Mycoplasmatota* was identified in 13 of the 40 groups, and phylum *Verrucomicrobiota* in 12 of the 40 groups, while the phyla *Chlamydiota*, *Chloroflexota* and *Spirochaetota* were found sporadically only in single groups. The phylum structure of bacterial communities is shown in Figure 2.

The following OTUs taxonomic distribution assigned them to 18 classes and 28 orders, divided into 64 families and represented by 140 genera. At the genus level, the dominant taxa, occurring in 97.5–100% of the control and experimental groups, were *Bacteroides* (belonging to the phylum *Bacteroidota*, class *Bacteroidia* order *Bacteroidales*, family *Bacteroidaceae*), as well as *Blautia*, *Butyricicoccus*, *Christensenella*, *Clostridium IV*, *Faecalibacterium*, *Fusicatenibacter*, *Oscillibacter*, *Romboutsia*, *Ruminococcus*, *Subdoligranulum*, related to phylum *Bacillota*, class *Clostridia.* A further 41 genera were found in at least 50% of the groups, showing them as resident members of the chicken caecum microbiome.

### 3.2. Bacterial Taxa Co-Occurrence in Chicken Cecal Microbiomes and Microbiome Patterns

As the next step, the co-occurrence of different taxa was explored and patterns of association between them were observed.

At the phylum level, highly significant correlation coefficients (*p* < 0.001) were shown between the five most abundant members of the cecal microbiomes (Figure 3). Multiple negative correlations were found between the phylum *Bacteroidota* on the one hand, and the phyla *Actinomycetota* (r = −0.553; *p* = 2.17 × 10^−4^), *Bacillota* (r = −0.988; *p* = 1 × 10^−20^), *Cyanobacteriota* (r = −0.559; *p* = 1.79 × 10^−4^), and *Thermodesulfobacteriota* (r = −0.577; *p* = 9.84 × 10^−5^) on the other hand. In turn, the phyla *Actinomycetota* and *Thermodesulfobacteriota*, as well as the phyla *Bacillota* and *Cyanobacteriota*, formed the co-occurrence clusters based on the following positive correlation coefficients: r = +0.886 (*p* = 2.78 × 10^−14^) and r = +0.586 (*p* = 7.05 × 10^−5^), respectively.

At the class level, this co-occurrence network structure was supported by significant correlation coefficients between the taxa that had dominated in the phyla described above. Class *Bacteroidia* (phylum *Bacteroidota*) demonstrated multiple negative correlations with the classes *Coriobacteriia*, belonging to the phylum *Actinomycetota* (r = −0.641; *p* = 8.29 × 10^−6^), *Clostridia* belonging to the phylum *Bacillota* (r = −0.866; *p* = 5.28 × 10^−13^), *Desulfovibrionia* belonging to the phylum *Thermodesulfobacteriota* (r = −0.577; *p* = 9.82 × 10^−5^), and a novel non-photosynthetic class *Melainabacteria* [29] related to the phylum *Cyanobacteriota* (r = −0.559; *p* = 1.79 × 10^−4^). In turn, classes *Coriobacteriia* and *Desulfovibrionia* (r = +0.898; *p* = 4.33 × 10^−15^), as well as *Clostridia* and *Melainabacteria* (r = +0.599; *p* = 4.45 × 10^−5^), strongly positively correlated each other, which formed the phyla co-occurrence network showed on Figure 3. 

In addition, when analyzing at the class level, the number of significant correlations (*p* < 0.001) between individual taxa increased to 17 and continued to grow at the order, family, and genus level. The co-occurrence network discovered at the genus level contained 5.88% of the possible connections (exclusively positive—723, with the exception of 25 negative correlation coefficients), which complicated routine analysis and required specialized graph visualization software.

Principal component analysis of inter-correlated quantitative taxonomic data presented orthogonal variables that integrally described the main patterns of chicken cecal microbiome compositions. The first weight vector F1 (eigenvalue 3.49; explained variance 28.94%) showed significant unrotated factor loadings (>0.700) for the four bacterial types. The highest factor loadings were found for *Bacillota* (+0.900) and *Bacteroidota* (−0.906), indicating opposite abundance of these phyla in cecal microbiomes. In addition, the F1 structure included the phyla *Actinomycetota* and *Thermodesulfobacteriota* (equal factor loading at 0.701), which was complemented by phylum *Cyanobacteriota* with a factor loading of 0.666 only, which was also added in the pattern due to the strong positive correlation with phylum *Bacillota* (see above). Thus, principal component analysis revealed the following two alternative patterns of the cecal microbiome: *Bacteroidota*-dominated F1(−) versus *Bacillota*-enriched F1(+), which co-occurred with members of the phyla *Actinomycetota*, *Cyanobacteriota*, and *Thermodesulfobacteriota*, and therefore designated as the *Bacillota* + ACT bacterial community. The second weight factor F2 (eigenvalue 2.12; explained variance 17.34%) extracted from the statistical data was not strictly associated with any taxon but allowed the distribution patterns of bacterial phyla to be displayed in spot maps (Figure 4A).

Subsequent principal coordinate analysis clearly assigned 38 of the 40 groups to F1(−) or F1(+) microbiomes as follows: 19 were scored as *Bacteroidota*-dominated bacterial communities and 19 were classified as *Bacillota* + ACT bacterial communities (Figure 4B).

### 3.3. Biodiversity in Chicken Cecal Microbiomes

The alpha diversity of chicken cecal microbiomes was assessed using the Simpson and Shannon indices calculated for the F1(−) and F1(+) groups separately. As shown in Figure 5A, the Simpson (1-D) evenness index, which measures the proportions of taxa in the microbial community, was 0.887 for the F1(+) related microbiomes, which was significantly higher (*p* = 1.07 × 10^−7^) than the 0.751 value for microbiomes associated with the F1(−) pattern. In turn, the Shannon index indicated both high taxon richness and equitability of distribution in the F1(+) related microbial communities at 2.860 versus a value of 2.021 for the F1(−) microbiomes (*p* = 6.08 × 10^−12^) (Figure 5B).

Thus, significant differences were observed in the alpha diversity indices between F1(−) and F1(+) groups, where *Bacillota*+ACT bacterial communities had higher phylogenetic diversity than *Bacteroidota*-dominated bacterial communities.

### 3.4. Influence of Chickens’ Cecal Microbiome Composition on the Poultry Farming Productivity

To evaluate the influence of the cecal microbiome composition on poultry farming productivity, the phylum and genera abundance were statistically compared with the collected zootechnical data at *p* < 0.001 significance and represented by the R^2^ coefficient of determination (Table 1). 

It has been shown that abundance of the phyla *Chlamydiota*, *Chloroflexota*, *Mycoplasmatota*, *Pseudomonadota*, *Spirochaetota* and *Verrucomicrobiota* in the chickens’ cecal microbiomes did not affect any productivity parameters, while some zootechnical parameters, including growth feed intake, average daily gain, and survival rate, do not change in the broiler chickens’ groups with various microbiome compositions.

Against this background, a complex positive effect on poultry productivity was revealed for the phyla *Bacillota*. High *Bacillota* abundance increased total feed intake (R^2^ = 0.383), especially through starter feed intake (R^2^ = 0.692) that resulted in a high final body weight by the 43rd day of feeding (R^2^ = 0.517). At the genus level, these effects were associated with the genera *Agathobaculum*, *Anaeromassilibacillus*, *Butyricicoccus*, *Fournierella*, *Ihubacter*, *Negativibacillus*, *Neglectibacter*, *Romboutsia* and *Roseburia*, which belong to the phylum *Bacillota* and are typically related to the class *Clostridia*. Significant R^2^ values were also established for the co-occurred phylum *Cyanobacteriota*, but their influence was weaker, did not include an increase in total feed intake, and no active genera belonging to this phylum were found.

The co-occured phyla *Actinomycetota* and *Thermodesulfobacteriota*, also included in the *Bacillota* + ACT pattern, showed a different effect on poultry productivity, changing the absolute gain parameter only as follows: R^2^ = 0.457 and R^2^ = 0.597, respectively. At the genus level, this effect was confirmed for *Rubneribacter* belonging to the phylum *Actinomycetota*, *and Bilophila* and *Mailhella*, both related to the phylum *Thermodesulfobacteriota.*

On the contrary, for the phylum *Bacteroidota*, a complex negative effect on poultry productivity was revealed. In groups where the chicken’s cecal microbiome was enriched with this taxon, a decrease in total feed intake (R^2^ = 0.419) was detected, especially due to the starter feed intake (R^2^ = 0.682), which led to a low final body weight by the 43rd day of feeding (R^2^ = 0.511). Among this phylum’s members, similar activity was most expressed in unclassified members of the family *Rikenellaceae*.

Surprisingly, a complex negative effect was also found in the *unclassified Bacteria*, which significantly decreased the starter feed intake (R^2^ = 0.394), increased the required amount of feed for 1 kg of live weight gain (R^2^ = 0.570), reduced absolute gain (R^2^ = 0.393), and resulted in low final body weight (R^2^ = 0.275).

### 3.5. Chicken Cecal Microbiome Pattern Effects on Zootechnical Parameters Distribution

The distribution of zootechnical datasets between the broiler chicken groups with F1(−) and F1(+) cecal microbiome patterns was analyzed by the Mann–Whitney U test at *p* < 0.001 significance (Table 2).

There were no differences in growth feed intake, feed conversion ratio per kg of body weight, average daily and absolute gain, as well as survival rate, while some zootechnical parameters depending on the *Bacillota* abundance (see above) were significantly higher in the *Bacillota* + ACT F1(+) microbiomes compared to the *Bacteroidota*-dominated F1(−) microbiomes. These differences included elevated starter feed intake (1971.00 g [1926.00–2196.40] vs. 1276.63 g [1095.96–1357.81]; *p* = 1.32 × 10^−5^), resulting in high total feed intake (4433.00 g [4231.33–4715.20] vs. 3984.15 g [3788.86–4177.52]; *p* = 3.76 × 10^−5^) and led to an increased final body weight (2931.67 g [2858.00–3288.00] vs. 2284.50 g [2214.00–2480.67]; *p* = 3.05 × 10^−7^). Interestingly, the phyla *Actinomycetota* and *Thermodesulfobacteriota* included in the *Bacillota* + ACT pattern did not affect the absolute gain parameter in this round of meta-analysis when comparing the F1(−) and F1(+) microbiomes.

### 3.6. Chicken Cecal Microbiomes Biodiversity and Poultry Farming Productivity

The dependence of chicken breeding zootechnical parameters on the Simpson (1-D) and Shannon biodiversity indices in their microbiomes was assessed using the R^2^ coefficient of determination, maintaining a significance threshold of *p* < 0.001.

High values of both indices corresponded to an increase in total feed intake, including starter feed intake, and high final body weight, which are similar to the effects of phylum *Bacillota* abundance and consistent with the concept of *Bacillota*-enriched microbiomes as the most diverse (see above). The Shannon index influence on starter feed intake was expressed more than the Simpson index (R^2^ = 0.776 vs. R^2^ = 0.542, respectively), which determined their effect on total feed intake (R^2^ = 0.547 vs. R^2^ = 0.340, respectively). These variations were confirmed in the chickens’ final body weight, where the effect of Shannon biodiversity (R^2^ = 0.665) was more intense than the Simpson biodiversity (R^2^ = 0.430). 

### 3.7. Cecal Microbiome and EPEF Associations

At the final meta-analysis round, zootechnical data were integrated with the European Production Efficiency Factor (EPEF) and compared with microbiome compositions and biodiversity indices using canonical correlation analysis, while their distribution between the F1(−) and F1(+) patterns was assessed according to the Mann–Whitney test.

A significant association was established between EPEF and *Bacillota* and *Bacteroidota* phyla abundance, which showed positive (r = +0.474; *p* = 2.03 × 10^−3^) and negative (r = −0.449; *p* = 3.70 × 10^−3^) correlations, respectively (Figure 6A,B). In addition, a positive correlation coefficient was shown for the phylum *Cyanobacteriota* (r = +0.371; *p* = 1.84 × 10^−2^), co-occurring with the phylum *Bacillota*, while a strong negative correlation was found between the FPEF value and the abundance of *unclassified Bacteria* (r = −0.706; *p* = 3.64 × 10^−7^), which corresponds to their negative influence on a number of zootechnical parameters (see above).

At the genus level, positive associations were found only for a few taxa, including *Butyricicoccus* (r = +0.566; *p* = 1.42 × 10^−4^), *Fournierella* (r = +0.575; *p* = 1.03 × 10^−4^), *Negativibacillus* (r = +0.504; *p* = 9.17 × 10^−4^), and *Neglectibacter* (r = +0.510; *p* = 7.74 × 10^−4^), which belong to the phylum *Bacillota*, class *Clostridia*, and affect several zootechnical parameters as described above. In turn, the negative EPEF associations with the *Bacteroidota* phylum abundance was confirmed by a significant negative correlation coefficient for unclassified members of the family *Rikenellaceae* (r = −0.549; *p* = 2.39 × 10^−4^).

Analysis of the EPEF distribution between chicken groups belonging to the F1(−) and F1(+) patterns confirmed higher values of 483.4 [449.02–533.9] for the *Bacillota* + ACT microbiomes versus 357.8 [324–462.4] for the *Bacteroidota*-dominated microbiomes, which was significant at *p* = 3.49 × 10^−4^ according to the Mann–Whitney U test (Table 2).

Increased biodiversity indices characteristic of *Bacillota*-enriched microbiomes and typical of the F1(+) pattern were also positively associated with high EPEF values. This association was more significant for the Shannon index, indicating high richness and equitability of bacterial taxa distribution in cecal microbiomes achieving increased productivity (r = 0.618; *p* = 2.10 × 10^−5^) (Figure 6C). The EPEF association with the Simpson (1-D) index, which measures the evenness of cecal microbial community, was also statistically significant (r = 0.498; *p* = 1.07 × 10^−3^) (Figure 6D) but less expressed.

## 4. Discussion

A meta-analysis of previously collected original data [14,15,16,17,18,19,20,21,22,23] allowed us to provide a detailed description of the chicken cecal microbiome at the phylum, class, order, family, and genus levels. After taxonomic classification at the phylum level, 11 taxa were found, dominated by *Bacillota* and *Bacteroidota*, together representing more than 90% of the bacterial community. These results are in good agreement with a previously published meta-analysis based on the largest data set of various publications describing the bacterial community using Illumina technology [7], which also reported the *Bacillota* and *Bacteroidota* phyla as most abundant in the chicken cecal microbiome. Subsequent rounds of taxonomic classification identified 18 bacterial classes, 28 orders, 64 families, and 140 genera, which is comparable to the meta-analysis of Wei et al. [5], who found 117 genera to be present in the chicken gut. Both our and a previous meta-analysis [7] showed the presence of *Faecalibacterium*, *Oscillibacter*, *Clostridium*, *Ruminococcus* and several other bacterial genera belonging to the phylum *Bacillota*, class *Clostridia*, in more than 50% of the samples, which is explained by their important role in food conversion through cellulose degradation and short-chain fatty acid synthesis. Thus, the current study is well concordant with previously published data based on sequences generated by the Illumina platform and strengthens the consensus around the composition of the chicken cecal microbiome.

The complete dataset in our study allowed us to develop ideas about the microbiome composition in the context of different taxa co-occurrence. First, *Bacteroidota* was found to have a strong negative association with several bacterial phyla, including *Actinomycetota*, *Bacillota*, *Cyanobacteriota*, and *Thermodesulfobacteriota*. The likely reason for this phenomenon is the ability of the *Bacteroidota* to synthesize a wide range of diffusible antimicrobial toxins called BSAPs (bacteroidales secreted antimicrobial proteins) [30], and thereby take part in interference competition in the host gut microbiomes. BSAP-1 is considered a key tool for dominance in vivo and as a result, the co-occurrence of the *Bacteroidota* phylum members producing BSAP-1 and sensitive strains belonging to other taxa is a rare event in the gut [31]. The second phenomenon was the discovery of the frequent co-occurrence of phyla *Bacillota* and *Cyanobacteriota*, as well as *Actinomycetota* and *Thermodesulfobacteriota*. This co-occurrence network structure was supported by pairwise combinations between some classes, orders, families, and genera belonging to these phyla; however, the reasons of inter-taxa associations in the bacterial community due to symbiosis, metabolic pathways, co-adhesion, etc., remain unclear and will be analyzed later using specialized software and methodological approaches.

Principal component analysis allowed us to formalize the cecal microbiome compositions, presenting it as two alternative F1(−) and F1(+) patterns with equal representation among all included in the meta-analysis groups. The F1(−) pattern was *Bacteroidota*-dominated and was characterized by decreased Simpson and Shannon indices values, indicating the biodiversity depletion of such microbiomes due to *Bacteroidetes* interference. In contrast, the F1(+) pattern represented a *Bacillota* + ACT bacterial community that was *Bacillota*-dominated, included members of the phyla *Actinomycetota*, *Cyanobacteriota*, and *Thermodesulfobacteriota*, and therefore showed increased biodiversity. The discovered patterns are closer to the concept of “enterotypes” first introduced by Arumugam et al. [32], which are also diet-driven and characterized by different digestive functions of the dominated bacterial taxa adequate for dietary habits. However, considering the “enterotypes” criticism [33], and since we obtained not a discrete, but a variable taxa distribution in the bacterial communities, in the present study, we used the term “pattern” as the most correct designation of the different compositions of cecal microbiomes. Thus, we can state a certain similarity of the F1(+) and F1(−) patterns with the PA1–3 chicken cecal enterotypes, demonstrating significant differences in the *Firmicutes/Bacteroidetes* (currently *Bacillota/Bacteroidota*) ratio, as well as differences in the alpha biodiversity, which was identified in a comparative study of 600 chickens belonging to two different breeds/lines from 60 farms with different farming practices [34]. Unfortunately, the chickens analyzed were often colonized by the *Campylobacter* genus, the most common zoonotic pathogen of foodborne diseases, which distorts microbial networks and does not allow their microbial networks to be considered “normal”. Notably, significantly different microbiomes dominated by *Firmicutes* and *Bacteroidetes* (currently *Bacillota-* and *Bacteroidota*-dominated) have also been described in broiler chickens under extensive production systems (EPS) or intensive production systems (IPS) [35], which showed low and high phylogenetic diversity, respectively. At the same time, the results of cited and present studies are not identical, since the F1(−) pattern did not include the phylum *Pseudomonadota* (formerly *Proteobacteria*), the F1(+) pattern included the ACT component, and there were additional differences from the bacterial taxa found to be at the core of the EPS and IPS microbiomes [11].

Consistent with the influential role of the chicken gut microbiome in nutrient absorption and metabolism [36], the cecal bacterial composition and microbiome patterns were compared with standardly collected zootechnical data including feed intake, weight gain, and survival rate. It has been shown that *Bacillota* + ACT F1(+) microbiomes, compared to F1(−) microbiomes enriched with *Bacteroidota*, have a positive effect on poultry productivity by elevated total feed intake (especially due to starter feed intake) and significantly increasing the chickens’ final body weight. Within the F1(+) pattern, these effects were predominantly determined by the phylum *Bacillota* abundance and were associated with the presence of several genera, typically belonging to the class *Clostridia*. Additionally, a positive association was found between poultry farming productivity and microbiome alpha diversity, which was higher in the F1(+) pattern compared to the F1(−) pattern. Having been integrated with the European Production Efficiency Factor (EPEF), poultry farming productivity confirmed the positive associations with F1(+) cecal microbiome pattern, the *Bacillota* phylum abundance, including *Butyricicoccus*, *Fournierella*, *Negativibacillus* and *Neglectibacter* genera, as well as the Shannon and Simpson (1-D) biodiversity indices.

It has previously been found that different enterotypes in broiler chickens play important roles in the degradation and utilization of plant polysaccharides, which may influence serum triglyceride levels and body fat deposition in chickens, but their effects on zootechnical parameters have not been considered [37]. In the study by Kers et al. [38], the cecal microbiota composition of broiler chickens have been reported in two robust community types associated with broiler health and performance, although complete descriptions of the microbiomes have not been provided. Against this background, we described for the first time the microbiome compositions that positively and negatively influence poultry farming productivity, which can be used to monitor feeding efficiency. Furthermore, we postulate the benefit of the *Bacillota* phyla abundance in the cecal microbiome, different members of which have specific activities relevant for improved feed efficiency and may be a source for novel probiotics [39,40]. Moreover, we provide insight into the importance of gut microbiome diversity, which underlies a wide range of metabolic pathways for efficient feed conversion and contributes to improved EPEF values.

Contrary to the reports of the possible beneficial effects of *Bacteroidota* on the enrichment of polysaccharide degradation pathways and the short-chain fatty acid production, as well as on the reduction of intestinal inflammation markers in one-week-old broiler chickens [41], the present meta-analysis showed that by the 43rd day of feeding, this phylum becomes zootechnically harmful, possibly due to the “egoistic” behavior and antagonism towards most other bacterial taxa in the cecal microbiome. 

An unexpected discovery was the finding of a significant negative effect on poultry farming productivity from a “*unclassified Bacteria*” group. According to the current concept, this category includes mislabeled taxa that are part of the extensive microbial diversity of the chicken gut microbiome identified by metagenomics [42], as well as representatives of potential “dark taxa” that have not yet been discovered [41]. It is assumed that these candidate taxa encode hundreds of unusual metabolic gene clusters and possess a distinctive functional capacity that might explain their elusive nature and bioactivity [43]. Since in the present meta-analysis “*unclassified Bacteria*” were associated with reduced starter feed intake, decreased absolute growth gain and low final body weight, we suspect the presence of new, previously unknown zootechnically harmful taxa, which requires further research to characterize them with the development of approaches to elimination from the chicken cecal microbiomes.

## 5. Conclusions

The meta-analysis provides new insights into the cecal microbiome compositions in regulating the growth performance of chicken and poultry farming productivity. Two alternative microbiome patterns are described and their associations with zootechnical parameters and EPEF values are established. These results lay the basis for the subsequent development of chicken growth-promoting feed supplementations, aimed at stimulation of beneficial and inhibition of harmful bacterial patterns, where the relevant metagenomic data can be a tool for their control and selection.

## Figures and Tables

**Figure 1 microorganisms-12-00747-f001:**
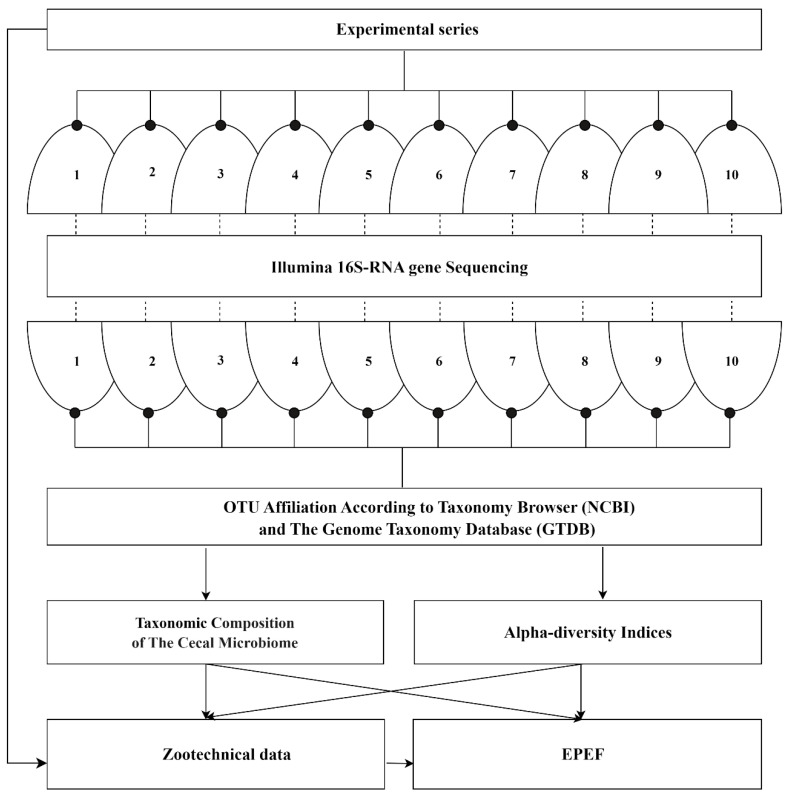
Meta-analysis design to characterize the cecal microbiome of broiler chickens and its association with poultry production. Designations 1–10 indicate the experimental series described in [14,15,16,17,18,19,20,21,22,23].

**Figure 2 microorganisms-12-00747-f002:**
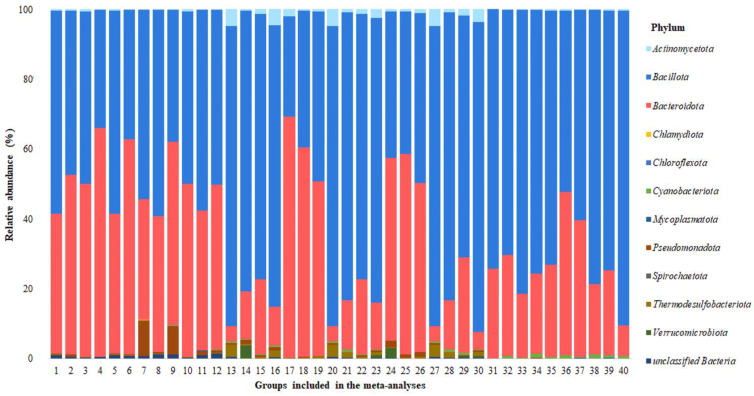
The relative abundance of bacterial phyla in cecal microbiomes.

**Figure 3 microorganisms-12-00747-f003:**
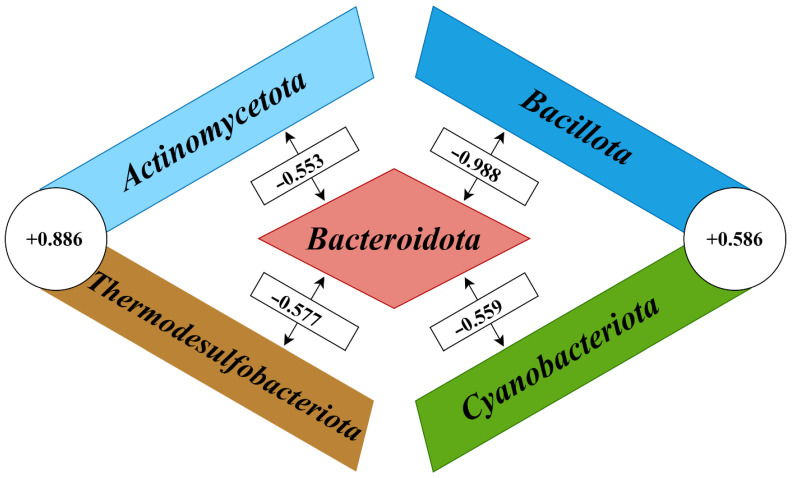
The phyla co-occurrence network in chicken cecal microbiomes.

**Figure 4 microorganisms-12-00747-f004:**
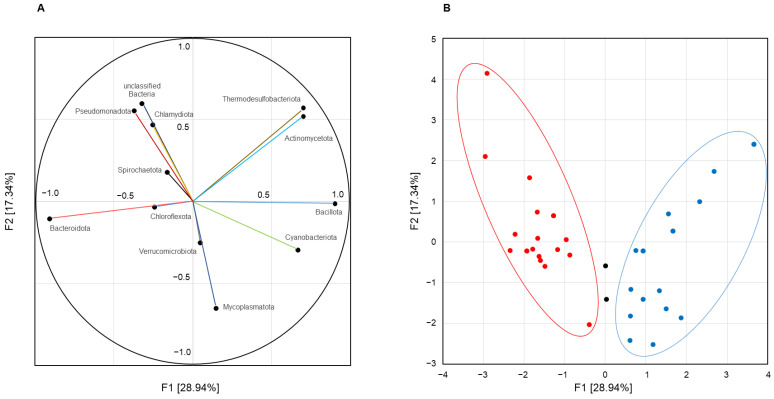
Alternative patterns in chicken cecal microbiomes assessed using principal component analysis. Part (**A**): Projection of bacterial phylum’s factor loadings onto the F1 × F2 factor plane. Part (**B**): Projection of microbiome compositions onto the F1 × F2 factor plane. *Bacteroidota*-dominated bacterial communities are on the left; *Bacillota* + ACT bacterial communities are on the right.

**Figure 5 microorganisms-12-00747-f005:**
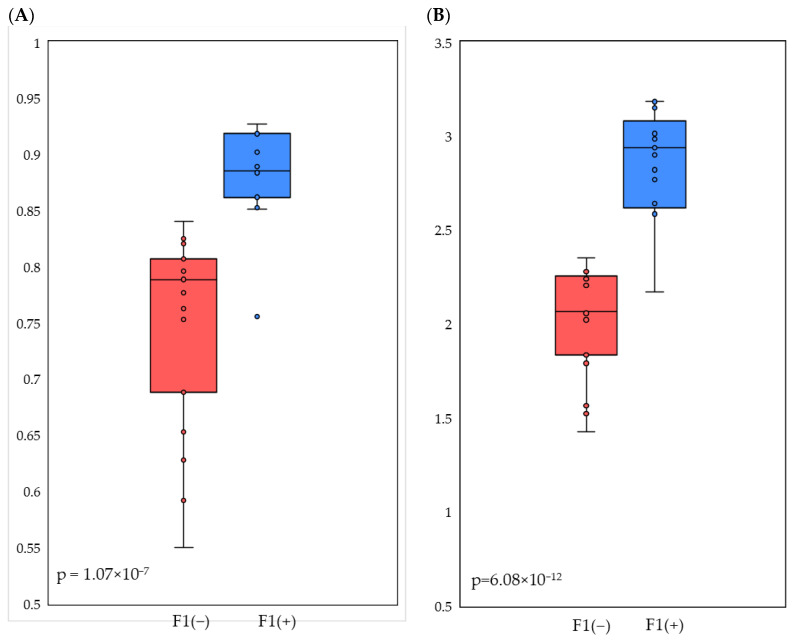
Simpson (1-D) (**A**) and Shannon (**B**) indices value distribution between chicken cecal microbiomes associated with F1(−) and F1(+) patterns.

**Figure 6 microorganisms-12-00747-f006:**
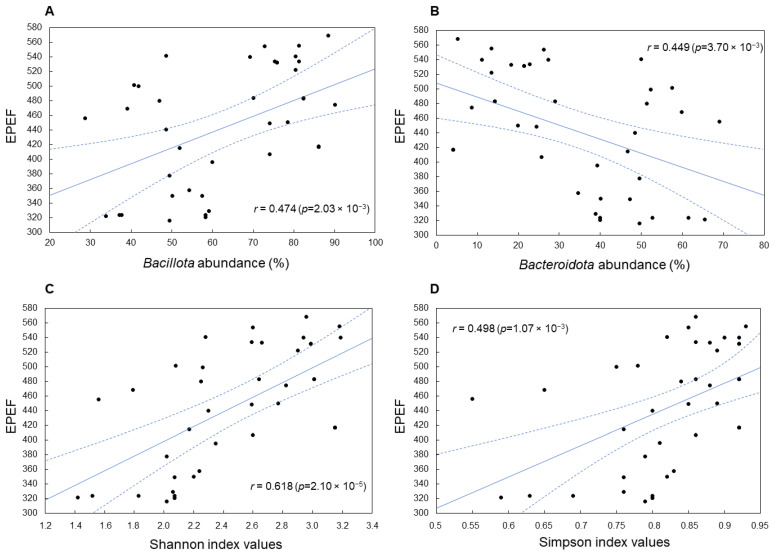
Associations between EPEF values and the abundance of *Bacillota* (**A**) and *Bacteroidota* (**B**) in chicken cecal microbiomes, as well as the Shannon (**C**) and Simpson (1-D) (**D**) biodiversity indices. The solid lines show observed regressions, dashed lines represent the 95% confidence regions.

**Table 1 microorganisms-12-00747-t001:** The influence of chicken cecal microbiome composition on poultry farming productivity, assessed by the coefficient of determination R^2^ *.

Zootehnical Parameters	Bacterial Phyla in Chicken’s Cecal Microbiomes **											
	A	B1	B2	C1	C2	C3	M	P	S	T	V	Unc
Starter feed intake		0.692	0.682			0.354						0.394
Growth feed intake												
Total feed intake		0.383	0.419									
Feed conversion ratio per kg of body weight												0.570
Average daily gain												
Absolute gain	0 . 457									0.597		0.393
Final weight		0.517	0.511			0.254						0.275
Survival rate												

* Black—positive influence. Red—negative influence. Empty cells (gray area)—no significant influence. ** Column designations: A—*Actinomycetota*; B1—*Bacillota*; B2—*Bacteroidota*; C1—*Chlamydiota*; C2—*Chloroflexota;* C3—*Cyanobacteriota*; M—*Mycoplasmatota*; P—*Pseudomonadota*; S—*Spirochaetota*; T—*Thermodesulfobacteriota*; V—*Verrucomicrobiota*; Unc—*unclassified Bacteria*.

**Table 2 microorganisms-12-00747-t002:** Distribution of zootechnical parameters between groups of broiler chickens with alternative cecal microbiome patterns.

Zootechnical Parameters	F1(−) Pattern	F1(+) Pattern
Starter feed intake (g)	1276.63 [1095.96–1357.81]	1971.00 * [1926.00–2196.40]
Growth feed intake (g)	2328.10 [2300.94–2440.61]	2464.67 [2377.75–2518.80]
Total feed intake (g)	3984.15 [3788.86–4177.52]	4433.00 * [4231.33–4715.20]
Feed conversion ratio per kg of body weight (kg)	1.86 [1.60–1.88]	1.68 [1.50–1.74]
Average daily gain (g/head/day)	72.95 [71.28–79.67]	76.46 [72.97–80.09]
Absolute gain (g/head)	2528.00 [2470.62–2749.12]	2320.00 [2273.60–2633.50]
Final weight (g)	2284.50 [2214.00–2480.67]	2931.67 * [2858.00–3288.00]
Survival rate (%)	98 [97–98]	97 [97–98]
EPEF	357.8 [324–462.4]	483.4 * [449.02–533.9]

Data are presented as median value [Q1–Q3]. * *p* < 0.001 between groups by Mann–Whitney U test.

## Data Availability

The sequencing raw data may be obtained upon request by e-mail at icis-ofrc@list.ru, belonging to the Institute for Cellular and Intracellular Symbiosis (ICIS) of the Ural Branch of the Russian Academy of Science (Orenburg, Russia).

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
