# Peer review of "Broiler Chicken Cecal Microbiome and Poultry Farming Productivity: A Meta-Analysis"

_microorganisms, 2024, doi:10.3390/microorganisms12040747_

Round 1

Reviewer 1 Report

Comments and Suggestions for Authors

About this study

The authors tried to identify the microbial community’s variants of the in chicken’s ceca and to ascertain the most significant association between the microbiome compositions and the poultry farming productivity.

This study is helpful study for those involved in poultry industry and farming.

Hard points:

Introduction: well written, introducing the reader in the general topic of this research.

M & M: is clearly presented, replicable and has a solid statistical analysis.

Results: probably is the best part of this manuscript, presenting clearly the obtained data, well associated with statistical significance of them (with the p values presented).

This study gives a detailed description of the chickens cecal microbiome at the phylum, class, order, family, and genus levels, a total of 11 taxa being found, where, Bacillota and Bacteroidota together represent more than 90% of the studied bacterial community.

Discussions: Are well conducted and based on actual information from 37 bibliographical titles, all linked to the topic.

Conclusion of this study brings relevant metagenomic data, which can be considered a an useful tool for the control and selection of the most suited chicken's growth-promoting feed supplementation.

What to improve:

Please add content in the introductory part about bacterial charge of gut especially about the microbial community, present in your areal.

Please develop more the “unclassified Bacteria” discussion, relevant for this study understanding.

Comments on the Quality of English Language

The English is fine, some minor issues, (typos) has to be corrected.

Author Response

Reviewer #1 comments:

  • Please add content in the introductory part about bacterial charge of gut especially about the microbial community, present in your area.

The authors agree with the reviewer that the gut microbial community is not constant and may vary between groups of chickens depending on many external factors.

There is some evidence that the microbiome composition and diversity may vary between chickens keeping in different types of indoor cage and cage-free systems [Adhikari, B., Jun, S.-R., Kwon, Y. M., Kiess, A. S., and Adhikari, P. (2020). Effects of housing types on cecal microbiota of two different strains of laying hens during the late production phase. Front. Vet. Sci. 7:331. doi: 10.3389/fvets.2020.00331], the use of different types of litter substrates [Wang, L., Lilburn, M., and Yu, Z. (2016). Intestinal microbiota of broiler chickens as affected by litter management regimens. Front. Microbiol. 7:593. doi: 10.3389/fmicb.2016.00593], and different diet supplementation [Zhou, J., Wu, S., Qi, G., Fu, Y., Wang, W., Zhang, H., et al. (2021). Dietary supplemental xylooligosaccharide modulates nutrient digestibility, intestinal morphology, and gut microbiota in laying hens. Anim. Nutr. 7, 152–162. doi: 10.1016/j.aninu.2020.05.010].

This content has been added to the “Introduction” part of the manuscript.

However, data on the specific gut microbial community present in broiler chickens from Russian agricultural area is still lacking, and further multicenter international studies are required.

  • Please develop more the “unclassified Bacteria” discussion, relevant for this study understanding.

According to the current concept, this category includes mislabeled taxa that are part of the extensive microbial diversity of the chicken gut microbiome identified by metagenomics [37], as well as representatives of potential “dark taxa” that have not yet been discovered [Liu, C., Du, MX., Abuduaini, R. et al. Enlightening the taxonomy darkness of human gut microbiomes with a cultured biobank. Microbiome 9, 119 (2021). https://doi.org/10.1186/s40168-021-01064-3]. It is assumed that these candidate taxa encode hundreds of unusual metabolic gene clusters and possess a distinctive functional capacity that might explain their elusive nature and bioactivity [Almeida, A., Mitchell, A.L., Boland, M. et al. A new genomic blueprint of the human gut microbiota. Nature 568, 499–504 (2019). https://doi.org/10.1038/s41586-019-0965-1].

This content has been added to the “Discussion” part of the manuscript.

  • The English is fine, some minor issues, (typos) has to be corrected.

Based on the reviewer's recommendations, we checked the manuscript and eliminated any revealed typos. The revisions marked in the text by green.

Reviewer 2 Report

Comments and Suggestions for Authors

The manuscript shows the results of a meta-analysis of the microbiome of the ceca of chickens subjected to different diets. The authors take the results of work done by themselves and already published. Therefore, the working methodology and the conditions of the animals are very similar and the data can be considered comparable. The article is interesting and conclusions are raised that are validated from the meta-analysis.

Some aspects must be improved for publication:

In the introduction the presentation of the problem is adequate. However, it would be interesting to raise some salient aspects of prior knowledge of the topic in greater depth.

In materials and methods, in addition to writing the conditions of the experiments that are included in the meta-analysis, the author must show them in a table would facilitating quick understanding of the study.

The results are very well graphed and tabulated, I only think that the , should be changed to . in the numbers shown.

Finally, although the discussion is well presented and allows important conclusions to be reached, there are many sentences in which comments and speculations are made without including references, for example lines 412-415; 449-460, among others. The incorporation of articles that support the opinions is necessary.

Author Response

Reviewer #2 comments:

  • In the introduction the presentation of the problem is adequate. However, it would be interesting to raise some salient aspects of prior knowledge of the topic in greater depth.

According the reviewer's recommendations, the “Introduction” section was supplemented as follows:

There is some evidence that the cecal microbiome composition and diversity may vary between chickens keeping in different types of indoor cage and cage-free systems [Adhikari, B., Jun, S.-R., Kwon, Y. M., Kiess, A. S., and Adhikari, P. (2020). Effects of housing types on cecal microbiota of two different strains of laying hens during the late production phase. Front. Vet. Sci. 7:331. doi: 10.3389/fvets.2020.00331], the use of different types of litter substrates [Wang, L., Lilburn, M., and Yu, Z. (2016). Intestinal microbiota of broiler chickens as affected by litter management regimens. Front. Microbiol. 7:593. doi: 10.3389/fmicb.2016.00593], and, especially, different diet supplementation [Zhou, J., Wu, S., Qi, G., Fu, Y., Wang, W., Zhang, H., et al. (2021). Dietary supplemental xylooligosaccharide modulates nutrient digestibility, intestinal morphology, and gut microbiota in laying hens. Anim. Nutr. 7, 152–162. doi: 10.1016/j.aninu.2020.05.010]. In turn, varieties in the cecal microbiome influenced nutrient metabolism [Yin Z, Ji S, Yang J, Guo W, Li Y, Ren Z, Yang X. Cecal Microbial Succession and Its Apparent Association with Nutrient Metabolism in Broiler Chickens. mSphere. 2023 Jun 22;8(3):e0061422. doi: 10.1128/msphere.00614-22. Epub 2023 Apr 5. PMID: 37017520; PMCID: PMC10286727.], including fat metabolism and fat deposition in broiler chickens [Chen Y, Akhtar M, Ma Z, Hu T, Liu Q, Pan H, Zhang X, Nafady AA, Ansari AR, Abdel-Kafy EM, Shi D, Liu H. Chicken cecal microbiota reduces abdominal fat deposition by regulating fat metabolism. NPJ Biofilms Microbiomes. 2023 May 30;9(1):28. doi: 10.1038/s41522-023-00390-8. PMID: 37253749; PMCID: PMC10229630.], and also were significant for growth promotion and individual differences in chicken body weight [Wang L, Zhang F, Li H, Yang S, Chen X, Long S, Yang S, Yang Y, Wang Z. Metabolic and inflammatory linkage of the chicken cecal microbiome to growth performance. Front Microbiol. 2023 Feb 23;14:1060458. doi: 10.3389/fmicb.2023.1060458. PMID: 36910194; PMCID: PMC9995838.]. However, a comprehensive analysis of the relationship between cecal microbiome composition and chicken’s productivity is still unavailable.

  • In materials and methods, in addition to writing the conditions of the experiments that are included in the meta-analysis, the author must show them in a table would facilitating quick understanding of the study.

I thank the reviewer for this recommendation to improve the understanding of the study. For this purpose, a description of the experimental conditions has been added to the Figure 1 legend. However, a separate additional table will duplicate Figure 1 and the text description and is therefore impractical.

  • The results are very well graphed and tabulated, I only think that the , should be changed to . in the numbers shown.

Done.

  • Finally, although the discussion is well presented and allows important conclusions to be reached, there are many sentences in which comments and speculations are made without including references, for example lines 412-415; 449-460, among others. The incorporation of articles that support the opinions is necessary.

Ability of the Bacteroidota phylum members to synthesize wide range of diffusible antimicrobial toxins called BSAPs (lines 412-415) supported by [Zafar, H.; Saier, M.H.Jr. Gut Bacteroides species in health and disease. Gut Microbes 2021, 13, 1-20. doi: 10.1080/19490976.2020.1848158.]. Previously reported data about microbiomes dominated by Firmicutes and Bacteroidetes (currently Bacillota- and Bacteroidota-dominated), as well as differences between cited and present studies (lines 449-460) supported by [Marcolla, C.S.; Ju, T.; Lantz, H.L.; Willing, B.P. Investigating the cecal microbiota of broilers raised in extensive and intensive production systems. Microbiol Spectr. 2023 11, e0235223. doi: 10.1128/spectrum.02352-23.].

All the revisions highlighted by green, and a revised manuscript resubmitted.
